# Understanding phenomenological experiences of autistic inertia using online community discourse
Tara Ward [1], Sonia Popazov[1], Jon Adams[1], Hayley Clapham[1], Wenn Lawson[1,2], Themis Karaminis [3] & Elizabeth Pellicano [1] ✉

The term 'inertia' refers to the seemingly common Autistic experience of remaining in a state of rest or a state of motion until there is some form of external intervention. While a heavily discussed phenomenon in the Autistic community, it has been scarcely acknowledged in the academic literature. The present study aimed to advance knowledge of Autistic inertia by analysing a large qualitative sample of naturalistic discourse on the topic from Autistic online communities on the social media platform, 'Reddit'. We identified 501 relevant posts shared between 2005 and 2023, including 9,955 comments. We analysed the posts using reflexive thematic analysis with an inductive approach. We identified four themes, centred on the "all or nothing" extremes of inertia (Theme 1), the range of factors that intersect with and exacerbate it (Theme 2), its joyful and often highly-disabling impacts (Theme 3), and the varied ways in which Reddit users manage it (Theme 4). Our findings corroborated those from existing interview-based studies and also uncovered additional insights, elaborating on 'the vicious cycle' of inertia, its fatiguing effects and its interaction with other commonly co-occurring conditions. We discuss these less-reported experiences and identify what we know - and are still yet to understand - about the key features of Autistic inertia.

There is growing appreciation that conventional ways of thinking about autism have constrained research and therefore our understanding of autism and what it means to be Autistic[1]. In recent years, both professional researchers[2,3] and community members[4] have highlighted the limitations of the conventional medical approach to autism, together with a failure to take Autistic testimony seriously[5]. This, it is suggested, has led to cases of 'undone science'[6] –or 'unknowing' existing knowledge about autism[7]. In this paper, we contend that 'Autistic inertia' is one such case. As such, we sought to build on the small handful of existing studies to deepen understanding of Autistic inertial experiences in an effort to develop a conceptual definition.

In 1999[8], Martijn Dekker reported on a particular issue "that we usually call 'inertia'", a seemingly common experience noted by members of Independent Living on the Autistic Spectrum, the online mailing list he managed. Dekker wrote:

> At first sight it looks like laziness: the inertial person has problems getting started with things, such as doing housework, filling in tax forms, or writing a paper for [the international conference held in 1999] Autism99, even if the motivation to do it is present… But on

the other hand, when the inertial person does manage to get started with something, it is hard to stop again; normal sleeping times are not observed and he gets irritable if interrupted because he is completely immersed in the activity that he finally managed to get started with.

Despite as "an almost universal trait" in Autistic people[8], it has received little attention by researchers. Only a few past studies have directly examined the first-hand inertial experiences of Autistic adults[9–11]. These studies have described Autistic inertia as the general tendency to maintain a single state, resulting in often-severe challenges in starting, stopping, or transitioning between tasks. Autistic participants reported on the often-disabling experiences of *inertial rest*, or challenges initiating everyday tasks. They have likened this to being 'frozen', 'stuck' (physically or 'in their mind'), or "having trouble getting the ball rolling"[12] (p. 7). Inertial rest can occur for a range of activities, including those that people find highly motivating, and is therefore not easily explained with reference to laziness, effort avoidance or lack of willpower. Participants have attributed these inertial rest difficulties to challenges with body awareness, including making a person's actions match their intentions, as

[1]Department of Clinical, Educational and Health Psychology, University College London, London, UK. [2]Curtin Autism Research Group, Curtin University, Perth, WA, Australia. [3]Department of Psychology and Neuroscience, City St George's University of London, London, UK. ✉e-mail: l.pellicano@ucl.ac.uk

well as difficulties in executive function, specifically in generating and/or executing mental plans.

These inertial rest experiences are akin to what we might think of when we use the term 'inert', as something or someone being motionless or unable to move. Yet, Autistic people also describe experiences of *inertial motion*, or challenges stopping tasks. Rapaport et al.'s[10] Autistic participants recounted experiences of being so deeply immersed in a task or a thought that they could not, or did not want to, stop, until interrupted. Some described being unable to stop tasks because they had entered a state of flow, emphasising the potential advantages of Autistic inertial motion, including high levels of productivity, a sense of control and a "a lot of joy"[10] (p. 7; see also[9,13]). Intriguingly, Autistic participants have reported these inertial rest and inertial motion experiences as "two sides of the same coin"—metaphorically similar to Newton's law of inertia: an object at rest remains at rest, and an object in motion remains in motion, until acted upon by an external force[14]. They have also emphasised the severity of these inertial challenges, akin to the feeling of living in "two extremes". As one of Rapaport et al.'s[10] participants put it, "there's nothing in between. I'm either go, go, go or can't move" (p. 5).

What is also clear from all first-hand accounts of Autistic inertia is its debilitating nature, with participants reporting profound and pervasive effects on all facets of life, including home, work, study, parenting and relationships. Even inertial motion can be so intense that it can be extremely difficult to shift to another task, which can have considerable negative repercussions[13]. These detrimental effects extend to Autistic people's well-being, especially their sense of self-worth, with many reporting feeling "full of shame" for being unable to do the same everyday things as others[10]. As such, it is unsurprising that Autistic inertia has been described as "the single most disabling part of being Autistic… it's a daily struggle"[10] (p. 6; also see[9]).

The qualitative studies conducted thus far have made progress in understanding Autistic people's phenomenological experiences of inertia and its impacts, using methods that have explicitly prompted participants to consider their experiences of inertial rest[9,10,12] and inertial motion[10]. The current study sought to complement these research methodologies by exploring naturalistic experiences of Autistic inertia. Specifically, we used data mining methods to extract a large volume of unprompted, text-based public posts about Autistic inertia from Autistic online communities on the social media platform Reddit (https://www.reddit.com), shared between 2005 and 2022.

We used Reddit because Reddit has no limit to the size of a post and includes a network of communities or moderated groups called 'subreddits' (e.g., r/autism), which allow like-minded individuals to connect over shared interests and topics (e.g., r/autism, r/aspergers). For example, a Reddit post could include a topic such as 'how do you cope with Autistic inertia?' and users can share as little or as much detail in the body section. Commenting users can also discuss original posts in detail, which reveal natural discourse on the topic. We analysed this discourse with the expectation that it would yield comparable experiences of Autistic inertia to those elicited in prompted and more constrained research environments[9,10,12] but, critically, also potentially uncover additional insights on what seems to be a core aspect of the Autistic experience, which, taken together, should bring us closer to a community-driven, formal definition of Autistic inertia.

## Methods
### Ethical issues
There are ongoing discussions about the ethical use of posts harvested from social media platforms, like X and Reddit, which are often perceived to be publicly-available data and often reported verbatim in research publications without seeking informed consent. Although information shared on Reddit is free for public use and Reddit's user agreement states that, unless deleted, public posts can be used for any purpose including research[15], users are largely unaware that their posts may be used in this way. Consistent with best-practice guidelines[16], we therefore treated users' Reddit posts as sensitive data. To protect users' confidentiality, we have replaced usernames with participant ID numbers and have replaced many direct quotations (or post excerpts) with paraphrased extracts. This approach protects user

anonymity and avoids exposing posts to a level of publicity beyond what users could reasonably expect[17]. We also refer to people as 'users' throughout, rather than 'participants', to acknowledge that they did not volunteer to participate in this research. This study received ethical approval from the Research Ethics Committee at University College London (project ID: 26315/001). This study was not pre-registered.

### Data extraction
We extracted a corpus of 5549 posts from the Reddit API (Application Programming Interface) using a set of search terms, since Reddit's inception (25/06/2005) to time of study (22/12/2023), within the following five subreddits: 'r\all', 'r\aspergers', 'r\autisminwomen', 'r\autism', 'r\autismtranslated'. Subreddit 'r\all' is a curated feed that shows the most popular posts across all subreddits, offering a broad view of content. The other four subreddits corresponded to specific, autism-related Reddit communities.

We were conscious of the search scope—using too many terms might result in redundant data and using too few could result in omission of information. After discussion within the team and piloting, we decided to be inclusive in our approach, so that we did not miss relevant posts that did not use the term 'inertia' simply because the user was unfamiliar with it or merely did not mention it. We therefore identified nine inertia-related search terms, including: 'inertia', 'hyperfocus', 'flow state', 'disruption', 'immersed task', 'interruption', 'monotropism', 'switching tasks' and 'transition tasks'. The search included morphological variants of these terms, such as third-person singular (e.g., "*switches tasks*") or progressive forms (e.g., "*switching tasks*"), or search terms with insertion of articles and prepositions (e.g., "*transition into a task*"). Furthermore, we included both jargon (e.g., 'inertia') and common phrases ('switching tasks') to ensure that data from users unaware of jargon were not excluded. Finally, each search term was combined with 'AND autis*' to limit the posts to discussions about autism/being Autistic. The posts and comments were generally lengthy, as users wrote about their experiences of inertia in depth.

### Screening strategy
Figure 1 illustrates the screening process. We performed the searches in December 2023 with PRAW (Python Reddit API Wrapper; https://praw.readthedocs.io/en/latest/index.html), a Python package that enables access to the platform's API and, effectively identifies posts and downloading them into CSV format (Microsoft Excel). The initial search yielded a total of 5549 posts, including 133,384 comments. This initial set of posts was pre-processed with a bespoke Python script to remove 1313 duplicates. The remaining 4236 posts were screened to determine eligibility. To be eligible, the content of posts needed to be (1) relevant to Autistic inertia; (2) authored by a user aged 18 or above; and (3) discuss the user's own Autistic experience. We therefore excluded (1) irrelevant posts, (2) posts that indicated the user was under 18 years (e.g., reference to school), and (3) posts that were written by a third-party (e.g., parent, researcher) or clearly stated the author was not Autistic. Importantly, as we note above, we adopted an inclusive approach to our search strategy to ensure that we did not miss relevant posts that used different terms or language. We therefore included inertia-related terms (like 'monotropism' and 'flow state') because users may have been using them to describe inertia-related experiences without using the term 'inertia', simply because they may not have been familiar with it. Posts containing those search terms were only analysed when the content itself aligned with challenges starting (inertial rest), stopping (inertial motion) and changing course. While users commonly disclosed their ages and stated their Autistic identity, we cannot be certain that all included posts met the criteria owing to limited information contained in the posts.

The first author screened all 4236 unique posts. Another researcher independently screened 10% of the posts, unaware of the first author's screening, yielding high (99%) agreement. Disagreements were resolved

**Fig. 1 | Flow diagram.** Figure outlines the screening process.

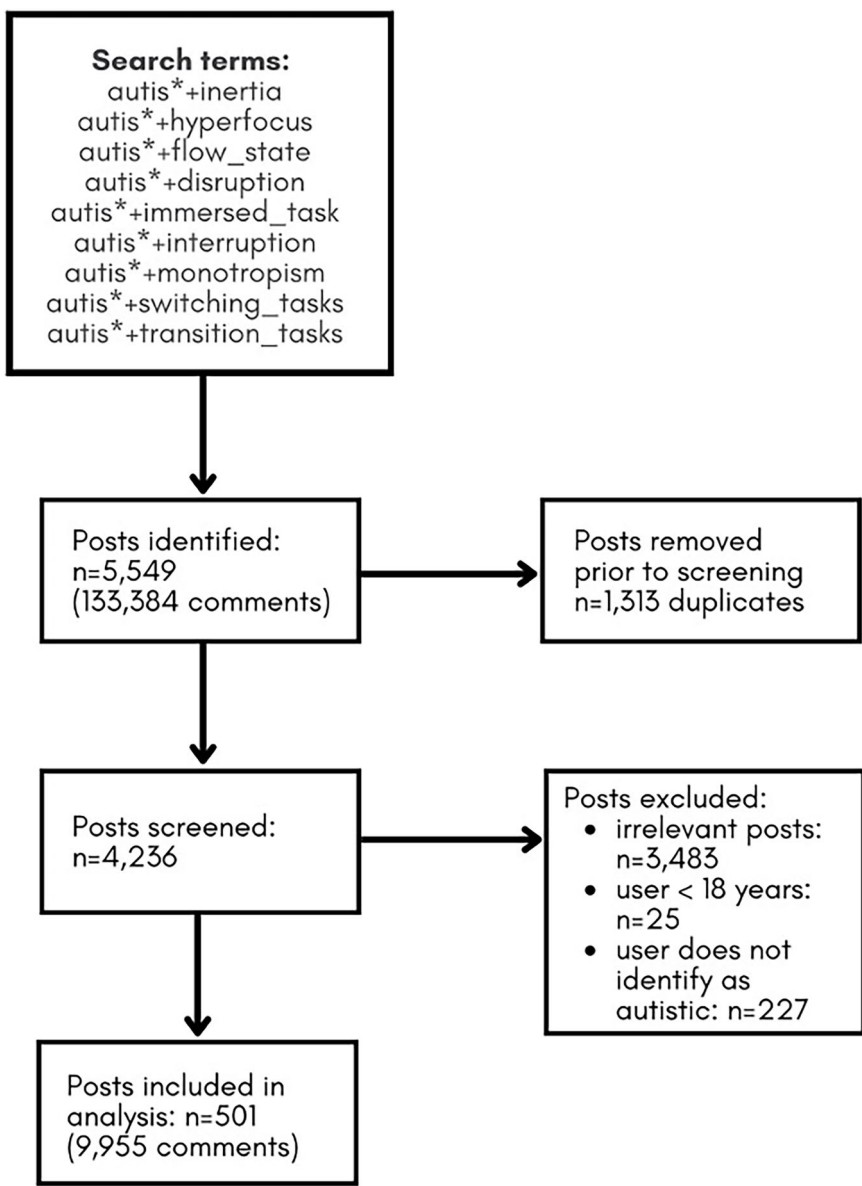

through discussion. Screening resulted in 501 relevant posts, including 9955 comments (M = 24 comments per post) (Fig. 1).

### Data analysis

Qualitative data, in the form of online discourse, were analysed using reflexive thematic analysis[18] within an essentialist framework, where we aimed to report participants' experienced realities. We used an inductive ('bottom-up') approach to identify key themes or patterns in the dataset, that is, without integrating the themes within any pre-existing coding schemes or preconceptions of the researchers.

All relevant posts and comments were transferred to NVivo Version 14 (Lumivero) and were re-read by the first author while applying reflexive notes. The data were then re-examined, applying codes to each post, in discussion with two co-authors. Codes were then aggregated into themes and subthemes. We adopted a semantic approach, where we sought to stay close to participants' language at the coding and theme-identification stages. The first author produced a thematic map, which was shared with, and discussed in detail by, the broader team. These discussions led to amendments in the thematic structure, theme and subtheme labels, until the team felt that all themes and subthemes were appropriate and showed a comprehensive picture of Reddit users' experiences of Autistic inertia.

### Positionality

Our team is a neurodiverse team consisting of Autistic advocates and researchers and non-Autistic researchers—therefore a mix of 'insider' and 'outsider' researchers. Our qualitative analysis was informed by this varied experience, as well as training in psychology, education, computer science, and creativity and mental health. It was also shaped by the social model of disability, a neurodiversity approach to research and practice, and our commitment to conducting autism research in partnership with Autistic people on issues prioritised by the community.

### Community involvement

Autistic scholars and advocates were actively involved in every stage of the research process from project inception (including being named on the original grant application) to choosing the research methods, designing the study (including identifying the search terms), analysing and interpreting the findings and commenting on the draft manuscript. Four of the team members (three Autistic, one non-Autistic) have worked together in a partnership for the past four years, to understand the nature of Autistic inertia. For this particular study, we brought three new researchers to the team (all non-autistic), both to offer additional expertise and to build their capacity in research adopting a participatory approach. The diverse nature

**Fig. 2 | Thematic map.** Figure illustrates identified themes and subthemes on Reddit users' experiences of Autistic inertia.

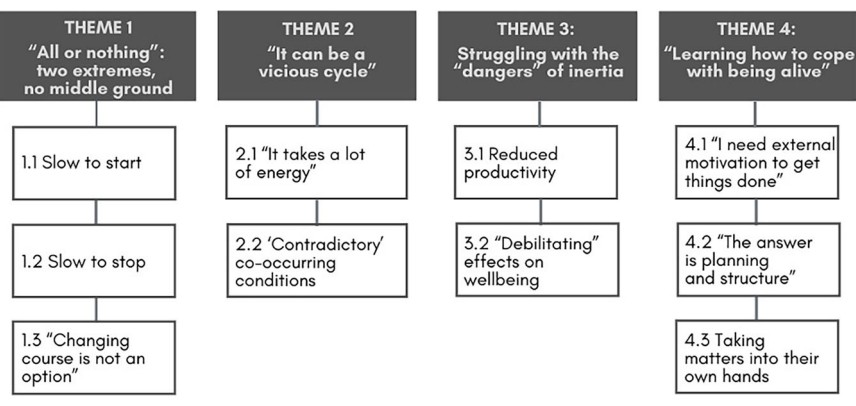

of our team and our participatory ways of working meant that we moved beyond neurotypical 'frames of analysis'[19] by looking at inertia – and of Reddit users' reported experiences of inertia—from different perspectives. We met regularly over Zoom to discuss ethical issues, the search terms prior to data extraction, eligibility criteria, and the analysis itself. The nature of our data (qualitative reports) mean that we often deferred to our Autistic partners' experiences to try to understand Reddit users' experiences.

### Reporting summary

Further information on research design is available in the Nature Portfolio Reporting Summary linked to this article.

## Results

We identified four themes (see Fig. 2), which are described in detail below. Themes are highlighted in bold and subthemes are italicised; illustrative quotes are attributed with the post ID.

### Theme 1: "All or nothing": two extremes, no middle ground

Overall, users framed experiences of Autistic inertia as wrestling with two extremes, rest and motion, and the challenges that arise from difficulties transitioning from one state to another.

**Subtheme 1.1: Slow to start**. Users frequently discussed their inability to "get started" (P19.6) on tasks—ranging from mundane domestic chores (doing the dishes, vacuuming the floor), to personal hygiene, work, interests and social activities. They described not being able to "find the trigger" to get moving (P19.4). For less-interesting tasks, users described how it felt like one's brain was being "dragged through nails to make it work" (P15.15). Starting was felt to be difficult no matter the nature of the task at hand; excitement, motivation and enjoyment did not seem to make starting any easier. P20.48 described this feeling as "lethargic" and taking any step felt like "too much", even if their "heart and soul disagrees and wants to continue" with the task in which they were currently engaged.

Users perceived this inability to get started to be caused by a metaphorical "barrier" (P4.14), which sometimes was said to manifest through a brain-body disconnect, or "mental block" (P9.2), whereby the person cannot physically get themselves to move despite the motivation to do so. This feeling was commonly described as being "stuck" (P7.14), "paralysed" (P7.16), or "chained in place" (P7.20). Some users said it was like being "trapped in a cage" of their own mind (P7.17), while others thought of it as a "fight" with their brain (P7.11). This inexplicable barrier was reported to be incredibly "frustrating" (P20.6), and for some it even "hurts" (P20.30).

Overwhelm could also cause a freeze response in users, when there were "multiple inputs" (P51.1) or when tasks had too many steps. One user described this as all the information "flooding" to their brain, which makes starting too "overwhelming" (P51.42), resulting in a state of "paralysis" (P51.54; P51.53). Having too many options of places from which to start could also lead to choice paralysis. In such situations, P12.2 described how

their "brain turns to mush and can't function"—often in response to domestic tasks, such as cooking and cleaning, which require the larger task to be broken down into smaller steps. Users commonly attributed this to executive function differences, and difficulties prioritising the order in which to complete tasks (P24.5).

**Subtheme 1.2: Slow to stop**. Conversely, the other extreme of inertia—difficulty stopping—was often perceived to be a positive experience, at least up to a point. Users described "intense moments of concentration" (P23.21) as flow states, periods in which "the stars align" (P23.12) and "everything feels right" (P23.31). Such states were reported to be "exhilarating" (P23.7), "satisfying" (P23.13), and "healing" (P23.16), which involved no thoughts, only "intuition and reflexes" (P23.29), leaving people feeling immensely productive, and "happy and fulfilled" (P23.26).

Yet, inertia did not always result in an enjoyable, productive flow state. Users described not having control over their own focus: "I can't break away!" (P69.1). Despite needing to eat, use the bathroom or complete other high-priority tasks, participants said they were unable to stop or shift until they finished what they started (P31.7). Some reported continuing a task despite it being physically harmful, occasionally resulting in "overuse injuries" from not taking a break for days (P11.5). Such "hyperfocus" could also lead to a loss of external and internal awareness (P31.25) and the neglect of bodily signals and losing track of time, leading to "physical distress" over not eating, drinking, or sleeping for days because of being "completely and wholly absorbed" (P23.5).

Users emphasised the single-focused nature of these states—wanting to focus only on one thing at a time. Parallel to being 'stuck' in inertial rest, users also described feeling "trapped" in motion (P46.10). Users described only being able to do one activity per day (P43.11), with housework, errands, and social life put aside because of an all-encompassing focus on work (P43.19) or interests (P37.42). P43.28 explained this inability to focus on multiple things at once as if their brain is a fast-processing computer with a small amount of RAM [memory], where running multiple programmes causes a shutdown.

**Subtheme 1.3: "Changing course is not an option"**. In addition to difficulties starting (inertial rest) and stopping (inertial motion), users described a third state: difficulties transitioning between states, which was felt to stem from the general tendency to maintain a single state of being (in this case, rest or motion). Difficulties changing course applied to a range of contexts. According to one user, transitioning from being dry to wet made it difficult to get up and shower (P37.53). Users also felt that transitioning from a state of doing something to stopping or switching tasks was just as challenging as transitioning from a state of "doing nothing" to "doing something" (P37.67). One user described shifting tasks as a task in itself: "You can't go from 'doing the dishes' to 'doing the laundry'; you have to go from 'doing the dishes' to 'shifting tasks' to 'doing the laundry'" (P46.12).

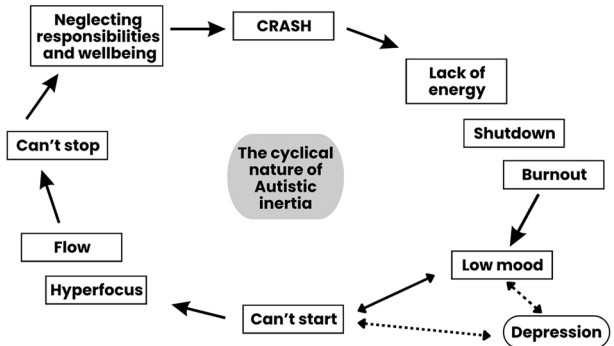

**Fig. 3 | The cyclic nature of Autistic inertia.** Figure illustrates the vicious cycle that can result from difficulties transitioning between states. Dotted arrows refer to the potentially compounding effects of low mood/energy and depression.

Difficulties transitioning between tasks/states seems to be "all or nothing" (P35.15), leading to a chain of events from "paralysis about getting started", to then "not being able to stop" once they had started (P35.19). P35.5 described taking a long time to get into the right gear, but once they do, they become "stuck" and unable to switch gears. This, in turn, makes starting "scary" because of the uncertainty around whether they could stop when needed (P35.19). On the other hand, some users avoided stopping because of the fear that the "motivation won't come back" (P35.27), resulting in inertial freeze.

Users also described immense distress of being interrupted when in motion. Interrupting a state of focus could "throw [them] completely off" (P37.8) or result in a state of "limbo" (P4.22). P35.6 explained this in terms of "driving a car with no breaks" - being unable to stop until no fuel is left, and "flying through the windshield and dying" if stopping too suddenly. Many attributed this to the immense energy cost from switching states, with the transition itself taking "an inordinate amount of time" (P37.69).

**Theme 2: "It can be a vicious cycle"**
Users repeatedly reported battling with Autistic inertia – and the range of factors that seemed to interact with and exacerbate it.

**Subtheme 2.1: "It takes a lot of energy".** Some users described having a "limited amount of energy" or "spoons" per day (P37.20) ["Spoons" here refers to "spoon theory," an analogy for fatigue and limited resources[20]]. They reported having to "keep check on energy levels" (P8.4) to prevent "over-using" their energy, which could lead to burnout and inertial rest. This meant they had to be selective about the tasks they complete and, sometimes, not engaging in interests or passions in exchange for having the energy to do "basic self-care" and vital tasks (P37.14).

Low energy was sometimes associated with low mood and depression, which users felt compounded inertia. They recounted being unable to "get out of bed entire days at a time" (P17.4), and could only "sit around, day after day" (P17.1). As a result, starting tasks became more difficult, making users feel "unproductive and useless" (P17.5), which further exacerbated the lack of energy and/or depression.

After periods of inactivity, some described garnering the motivation to have a few productive "high-energy days" to offset neglecting everyday tasks. However, once these tasks were started, users often reported getting into a rhythm and ended up "overexerting" themselves (P37.55). Continual overexertion could lead to "overload" and "shutdown" (P8.7), a state in which users "pay" for being productive (P37.16), and made doing anything that requires a modicum of energy "extremely difficult" (P8.14). Users reported this made them more prone to burnout. Figure 3 depicts the cyclical nature of inertia.

Users' posts also revealed that repeated cycles of inertia were perceived to result in loss of skills ("skill regression"), leading to more severe inertial

cycles in older Autistic adults. Some reported noticing their inertia "getting worse as [they] age" (P53.3), and were "losing the battle with autistic inertia" (P53.4). These age-related changes were often attributed to diminished executive function and lower energy levels compared to when they were younger.

**Subtheme 2.2: 'Contradictory' co-occurring conditions.** This vicious cycle was further compounded by many users' co-occurring conditions, which reportedly influenced the extent and nature of their inertial experiences. For users who identified as AuDHD (Autistic people with co-occurring ADHD), characteristically ADHD features like difficulties focusing and the need for novelty contradicted their Autistic need for routine. While routines led to under-stimulation, making it more difficult to start tasks (P1.18), over-stimulation and lack of structure resulted in overwhelm (P1.43) and "shutdown" (P1.46). P1.47 described this as "two sides going against each other in [their] head". Some also described a strong sense of switching between an Autistic "I need to do this" mode to an ADHD "I can't do the same thing for more than 30 min" mode (P75.1). ADHD paralysis/hyperfocus and Autistic inertia—as they referred to it—both resulted in shutdown during overwhelm and included difficulty switching focus and breaking away once starting.

Other conditions reportedly affected inertia, too. Pathological demand avoidance (PDA) was cited as making initiating and stopping tasks difficult. Although PDA is typically regarded as an Autistic profile where a person is resistant to other people's demands, users described also being resistant to their own wants and needs—that is, their own 'demands'. This made the things they wanted to do to seem like "insurmountable demands" (P50.10). Some described having to "manipulate" (P50.9) or "argue with" themselves (P50.21) into doing tasks, because simply wanting to do them was not enough (P50.9).

Though less frequently mentioned, obsessive compulsive disorder (OCD) interacted with Autistic inertia for some users. Those with co-occurring OCD reported experiencing more severe inertial motion, whereby being productive could "feed the OCD beast" leading to an inability to sleep or complete basic tasks until the project of focus is completed (P46.1).

**Theme 3: Struggling with the "dangers" of inertia**
**Subtheme 3.1: Reduced productivity.** Users described how inertial motion could result in immense productivity and "accomplishment" (P79.10) when the conditions were right, but the "boom-and-bust" cycles (P30.5) of inertia (Fig. 3) could also adversely affect productivity. Inertia seemed to take a particularly large toll on users who were studying, with many citing that getting into "school mode" was a particularly difficult start (P18.3). While some reported receiving poor grades, others discussed having dropped out of university or college because of inertia (P18.10). Users also reported how inertia affected productivity at work, especially their inability to turn on "work mode" (P59.2). On the other side of this "double-edged knife", once having started a task, users would often "lose sight of other more important things". This resulted in poor job performance, and in extreme circumstances, job loss (P38.7).

**Subtheme 3.2: "Debilitating" effects on wellbeing.** The consequences of inertia were not solely related to work and study; users also reported that it often had detrimental consequences on their health and wellbeing. They often neglected basic self-care. Difficulties doing chores resulted in living spaces being a "mess" (P19.11). Personal hygiene suffered, as they could not "shower or brush [their] teeth", which they felt was "debilitating" (P15.7). Eating was also reported to be difficult, especially when cooking required many steps. Some reported being underweight as they resorted to eating non-nutritious snacks, which required less energy (P25.10). Yet, others, once they started eating, found it "nearly impossible to stop" (P21.1). Sleep hygiene also suffered, as users reported being unable to "put the activity down and go to bed" once started (P59.21).

Consequently, inertia could leave users feeling "guilty and ashamed" because they could not function as well as others (P27.12). Many blamed

themselves for not being productive or functional, calling themselves "lazy and stupid" (P27.24). This resulted in lower self-esteem and feeling like they were a "huge burden to live with" (P27.37). Some cited how inertia had ruined their relationships. Taking the step to contact people was reported to be difficult, causing friends or family to have been "upset" and "accuse[d] [the user] of giving them the cold shoulder" (P58.8).

### Theme 4: "Learning how to cope with being alive"
**Subtheme 4.1: "I need external motivation to get things done".** Although inertia could be challenging, users repeatedly posted on ways they sought to "lean into" (P72.1) their cognitive style and manage their inertia. They reported wanting to be productive but also wanting to do it in a balanced, less harmful way (P77.1). One way was through "body-doubling"; having someone present when doing a difficult activity, like studying at home (P6.3). Many found it helpful to be given instructions or prompts by other people. Some relied on others to the point where they "freeze up and fail to function" without help (P6.34). Others reported "thriving well" in situations where they can be "held accountable" (P6.30) through external deadlines. Several users had even reportedly trained service animals to help keep inertia in check by interrupting hyperfocus or prompting movement. However, they also noted how prompts and interruptions could be "annoying and jarring" (P25.28), which meant they often traded peace with productivity.

**Subtheme 4.2: "The answer is planning and structure".** Users also reported relying on strict routines and planning to stop them from slipping into inertia. Many meticulously planned their day using programmes and applications to set reminders and prompts throughout the day. Detailed planning and breaking down tasks reportedly helped with overwhelm and uncertainty that could lead to choice paralysis. Many reported finding it helpful to set a "middle space" between timers to help shift between tasks, making transitioning between states less daunting (P41). However, sudden alarms could create "a surge of anxiety" in some (P37.11), which made transitioning difficult. Breaking tasks into too many small steps could also create overwhelm (P51.34).

**Subtheme 4.3: Taking matters into their own hands.** Notably, users repeatedly discussed their negative interactions with medical professionals, including both a lack of understanding of their day-to-day challenges and not being taken seriously. P37.27 recounted how their therapist "did not know what [they] were talking about" when they opened up about Autistic inertia. Many also wrote of being medicated for co-occurring ADHD. Some AuDHD users relied on ADHD medication to "get something done", as it led to "increased focus and executive functioning" (P41.27). However, users also described how ADHD stimulants could cause "detrimental" (P41.1) hyperfocus to the point that it becomes "death focus" (P41.5). As a result, many AuDHD users felt they could not "take full advantage of [their] medication's effects" (P41.2).

When professionals were unable to help, many resorted to self-medicating to cope with inertia. Self-medicating with drugs such as marijuana was said to help "quiet [the] brain enough to focus", making initiating tasks seem less overwhelming (P55.10). Others reported using psychedelics to increase energy, with P55.24 reporting feeling like there is "nothing blocking [them] from getting going". While self-medicating with drugs had some perceived benefits on inertial rest, it did not, however, seem to prevent users' reported challenges with inertial motion.

## Discussion
Here, we explored the lived experiences of Autistic inertia as shared by online Autistic community members on the social media platform, Reddit. Our research methods complemented existing interview-based research[9,10,12], providing rich and nuanced insights into naturally-occurring experiences with a larger sample of users across a long period of time. The findings confirmed those from existing research regarding the phenomenological experience of inertia, particularly reports of living in

"two extremes" (Theme 1), and both its joyful and disabling impacts (Theme 3). Critically, however, our findings also revealed insights into less-reported experiences, including 'the vicious cycle' of inertia, its fatiguing effects and the factors that exacerbate it (Theme 2), as well as the varied ways in which people manage its benefits and pitfalls (Theme 4). Below, we focus our discussion on these less explored findings, before bringing them together with those of existing research to identify what we know - and are still yet to understand - about the key features of Autistic inertia.

One key finding relates to users' reports of a vicious cycle that arises from difficulties transitioning between states—and the way in which these difficulties intersect with co-occurring conditions. Users' reports suggested an inertial loop that appears to affect—and be affected by - co-occurring depression (Fig. 3). They described how overwhelm or shame associated with inertial rest could lead to low mood or depressive symptoms, making it even harder to start an activity, and perpetuating a loop of inactivity and negative emotions. When users reported being able to 'break free' of the inertial loop and start a task, either through considerable effort or external intervention, they then felt propelled into the other 'extreme', involving often-prolonged periods of "high-energy" inertial motion (hyperfocus or flow). Although such periods could be joyful in the moment, users described how transitioning out of them could take its toll, resulting in neglecting basic self-care and other responsibilities, significant energy depletion and, eventually, a 'crash' (Fig. 3), which could result in fatigue, shutdown and, in some instances, burnout. The fatiguing effects of inertia made it often exceedingly difficult to act, further perpetuating the cycle.

The current findings build on those from previous studies[9,10] to suggest that negative emotions and co-occurring depression—unfortunately common in Autistic people[21]—may play a role in Autistic inertia - and vice versa, as it is possible that inertia could be a maintenance factor in depression. Understanding the precise nature of this relationship and identifying interventions focused on disrupting the inertial loop are important avenues for future research.

Within the context of this vicious cycle, users repeatedly recounted the fatiguing effects of inertia. These energy costs were briefly reported in previous research, with Rapaport et al.'s[10] participants likening the energy costs involved in transitioning between tasks to Newton's law of inertia: "When something is stopped, it requires a certain amount of energy to get it started. And then when it's started, it requires a certain amount of energy to get it stopped again" (p. 5). Inertia's potentially fatiguing effects were described in more detail in our dataset. Such energy costs could be explained, at least in part, by monotropism[22–24]—the notion that Autistic people tend to focus intensely on one or a few interests or tasks at any time, compared with non-autistic people, who are thought to have a polytropic tendency, or multiple interests aroused at any time. The monotropic state, or "hyper-awareness within the attention tunnel"[23] (p. 142), is thought to attract most of a person's processing resources, leaving fewer resources for other processes, including tasks outside of the current attentional tunnel.

Intriguingly, the energy costs associated with Autistic inertia and a monotropic mind more broadly may relate to Autistic people's fluctuating performance (skill regression) on seemingly ordinary tasks. Some days, a person may be able to accomplish tasks with ease, while on other days, they may find the same tasks insurmountable and find even basic tasks difficult, making people feel "unproductive and useless" and leading to overwhelm. Although there is much discussion of fatigue and energy management in the Autistic community, it is rarely discussed in the academic literature[25,26], making it a worthy avenue for future research.

The exception to this, however, is its integral role in Autistic burnout. For Autistic people, burnout is a state of intense mental, physical, and emotional exhaustion experienced by autistic people, which is understood to be the result of the energy-draining effects of masking and living in an unaccommodating neurotypical world[27–29]. While burnout is typically understood to be conceptually distinct from inertia[12], our users felt that inertia and burnout could be related, in one way specifically. For them, repeated episodes of inertia were perceived to be a possible precursor to

Autistic burnout. It is also possible, however, that burnout could exacerbate inertia—or both. Given the dearth of research on these constructs, future research efforts should focus on understanding the extent and nature of the relationship between Autistic inertia and Autistic burnout—and role of Autistic fatigue and energy depletion in both states.

Our findings also revealed that users' Autistic inertia was felt to be heavily influenced by other co-occurring conditions common in Autistic people, including PDA, or 'Persistent Drive for Autonomy' as it is sometimes referred to in the community[30], ADHD and, to a lesser extent, OCD. PDA, considered by some a specific profile of autism, describes a strong need to question and resist demands arising from an overwhelming need for control[31]. Reddit users described how their PDA traits worsened their inertia, particularly due to a strong resistance to what they saw as their own demands - even with tasks they genuinely wanted or needed to do. Difficulties dealing with internal demands are discussed by Autistic people[32], but have, once again, rarely been described in the literature. Users also referred repeatedly to how their co-occurring ADHD can affect how they experience inertia. They reported how being ADHD can cause them to seek a variety of tasks (related to novelty-seeking), which can conflict with their Autistic desire for predictability and monotropic (single-focused) nature, often making it harder for them to manage Autistic inertia. Understanding precisely how co-occurring PDA and ADHD traits might influence Autistic inertia requires further examination.

Beyond influencing factors, our users discussed at length the varied ways in which they manage inertia, which might be related partly to the opportunities afforded by online communities like Reddit to share experiences, offer support and connect with others facing similar issues[17,33]. Like previous studies[9,10], we found that external intervention was often felt to be effective in breaking out of inertia (if not sometimes also distressing), as well as structured routines and detailed planning. However, when these strategies failed to help, some users reported relying on their ADHD medication or turning to self-medicating with recreational drugs. This finding might seem surprising given that substance use is thought to be low in Autistic people[34,35]. It is, however, consistent with a recent large-scale study, which examined self-reported substance use in anonymised Autistic and non-autistic adults using quantitative and qualitative methods[36]. While quantitative analysis showed decreased use of recreational drug in Autistic people, qualitative analysis revealed that Autistic people were nearly nine times *more* likely than their non-autistic peers to report self-medicating (with substances such as marijuana, cocaine and amphetamines) for mental health issues. This increased risk may be further compounded by co-occurring ADHD[37].

Interestingly, users in the current study identified reasons for turning to self-medication to manage unwanted challenges (including inertia), namely healthcare professionals' lack of knowledge about autism/inertia and their dismissal of users' concerns. It is well established that healthcare professionals' outdated and sometimes prejudicial perspectives can hinder their ability to engage with Autistic people and provide appropriate care[38–42], and can even cause Autistic people to avoid seeking formal healthcare support[39]. Much more needs to be done to provide Autistic people with access to high-quality healthcare that is respectful of, and attentive to, their specific needs. We also need greater understanding of the potential role of pharmacological interventions in remediating the disabling effects of Autistic inertia.

These findings also highlight the value of utilising a range of methodologies when investigating cases of undone science, like inertia. For Weir et al. [36], an anonymised open-ended survey led to revelatory responses on substance use. Similarly, the (pseudo)anonymity inherent in Reddit's structure, the presence of moderators, and the opportunity to write lengthy posts and comments also lends itself to an online space that allows for in-depth personal discussions and disclosures[43], as attested by users in this study. Analysing Reddit data provided access to a naturally-occurring dataset, which allowed for more forthright sharing of experiences.

### Toward a definition of Autistic inertia
Drawing together the findings from the current study and previous ones[9–12] brings us closer to a formal definition of Autistic inertia (Fig. 4), centred on the general tendency to maintain a single state of being. Autistic young people and adults' first-hand accounts—across methodologies—are highly consistent regarding phenomenology. Autistic people reportedly wrestle with two "extreme" states, feel physically and mentally "stuck" in those states, and experience substantial challenges transitioning between them. They are also resoundingly clear on its impacts, that both inertial rest and inertial motion can be highly debilitating, but also give rise to positive states (flow), which can be joyful and support wellbeing[10]. It appears to be distinct from other related issues, such as procrastination, depression, catatonia and burnout, although all are important to consider in differential diagnosis. Its precise (causal) relationship with other conceptually-similar constructs, such as differences in executive function, movement/coordination or predictive coding, is unclear[9–11].

A community-driven consensus definition of Autistic inertia is essential to inform the design and development of tools to measure it. The construction of a well-validated questionnaire, for example, would allow researchers to differentiate Autistic inertia from other conditions, delineate its underlying mechanisms, understand the extent and nature of its effects on everyday functioning and wellbeing, and develop effective ways to help Autistic people manage the more disabling aspects of inertia. Before that can happen, however, several outstanding issues warrant further exploration.

The first relates to the temporal qualities of Autistic inertia. While some findings have suggested that Autistic inertia "pervades every single day" and could occur "all the time, with almost all tasks"[10] (p. 6), it is also described as being episodic in nature[9,11] and dependent on context[10]. Neither the frequency and duration of the inertial episodes (or cycles) nor their triggers are currently well understood.

The second unresolved issue relates to whether inertia is a uniquely Autistic experience. There is much discussion of concepts like flow, hyperfocus, task paralysis, task switching and even inertia itself in the ADHD community and in the literature itself[44,45], raising ongoing queries about (a) the inertial experiences of the substantial proportion of Autistic people who are also ADHD[46] (emphasised in the current findings) and (b) its potential transdiagnostic status - that is, whether inertial rest/motion is shared among people with other forms of neurodivergence.

Finally, while stressors and external (contextual) demands can exacerbate experiences of Autistic inertia, the full range of factors that influence, or are influenced by, Autistic inertia requires greater exploration. We do not know, for example, whether Autistic inertia is affected by hormonal issues (menstruation, menopause), which are reported to amplify sensory overwhelm and executive function differences[47]. We also do not understand in depth how interoception—an understanding of your body's internal senses (e.g., of hunger or when to go to the toilet)—relates to inertial rest and motion, especially since many first-hand inertial accounts emphasise feelings of challenges with bodily awareness. While these issues will be teased apart once there is an agreed definition and valid and reliable ways of measuring Autistic inertia, they are still ripe for investigation with rich qualitative methods.

### Limitations
This study has several limitations. First, Reddit's privacy regulations meant that we were unable to report users' demographic data, including their racial/ethnic identities, ages, locations or diagnoses. While the written nature of the posts may have encouraged or included some Autistic users who are often excluded from interview-based studies (e.g., those who do not or prefer not to use spoken communication), we cannot be sure of the representativeness of our sample. Also, although users often reported their diagnosis/identity, it was not always possible to know whether a user was Autistic. Second, due to the technical characteristics of the Python PRAW library and the Reddit API, we cannot be certain how exhaustive the results of the keyword-based search were or whether some users, posts or comments were missed. Third, questions have been raised about the quality of information on social media —a place where users are more able to misrepresent themselves to spread misinformation[48]. While this might be true for some groups, content analysis of autism forums, including on Reddit, has revealed "minimal hostility and

**Fig. 4 | Preliminary defined criteria for Autistic inertia.** Figure describes the key features of Autistic inertia and those that distinguish it from other, conceptually-similar issues.

**Autistic inertia** refers to the general tendency to maintain a single state of being, which results in challenges starting (inertial rest), stopping (inertial motion) and changing course. These states can occur regardless of willpower, and yield different consequences (some negative, some positive).

Inertia can be experienced as "living 'in extremes'", with a feeling of being physically and/or cognitively 'stuck' and challenges with bodily awareness (mismatch between intentions and actions).

Transitioning out of inertial rest or inertial motion can be debilitating, leading to energy depletion, highly variable functional capacity, overwhelm and, eventually, burnout.

Prompts or 'interventions' from a person or other external 'force' (e.g., a timer, deadline), routines and structure, and breaking down smaller, more easy-to-complete tasks, can facilitate action. Mental health difficulties and stress can exacerbate inertia. The influencing effects of co-occurring conditions (ADHD, PDA) are not well understood.

**Distinguishing Autistic inertia:**
- *Procrastination:* Difficulties starting may be explained by procrastination (i.e., knowing that one needs to perform a task, but not being motivating to do it), but it cannot explain difficulties starting regarding highly motivating tasks.
- *Depression:* Some features of Autistic inertia (e.g., feeling cognitively and emotionally stuck, low energy) overlap with depressive characteristics, but others (difficulties stopping, task immersion and flow experiences) do not. Inertia can, however, affect and be affected by, co-occurring depression.
- *Catatonia:* Although Autistic inertia has several characteristics in common with catatonia (e.g., freezing and getting stuck[50]), Autistic people report a broader range of body awareness differences, including feeling a 'mind-body disconnect'.
- *Burnout:* Autistic burnout has been defined as a state of profound physical, mental and emotional exhaustion driven by the stress of masking and living in an unaccommodating neurotypical world. The precise relationship between Autistic inertia and Autistic burnout requires more research.

virtually no misinformation"[49] (p. 11), with such forums shown to contain high-value insights for clinicians and scientists alike. Fourth, unlike traditional interview-based methods, social media analysis does not allow for follow-up with any users, potentially limiting the depth of information provided. That said, Reddit's comment function allows for substantial discussion by other users, which is not dissimilar from the discussion generated within a focus group, albeit in (pseudo)anonymised form.

## Conclusions

Our analysis successfully corroborated the findings of inertial rest and motion from existing interview-based research[9,10,12,13] in a large sample of online naturalistic discourse via Reddit. Critically, it also raised insights not previously explored in depth. The cyclical nature of Autistic inertia and its interaction with other commonly co-occurring conditions is complex and not well understood. Additional research is needed to differentiate between other conditions, as well as identify interventions that can help manage the debilitating sides of Autistic inertia, while also promoting the conditions that result in positive flow experiences and thus foster Autistic wellbeing.

## Data availability

In accordance with our ethical approval, we have not deposited the de-identified data because doing so would make users included in our research identifiable and subject to greater publicity than they may have reasonably expected. Researchers, however, can apply the openly

available script to Reddit queries, which are openly available online. De-identified qualitative data are available from the corresponding author upon request.

## Code availability

The script used to retrieve the raw Reddit data used in this study is available on the Open Science Framework and on request to the authors (https://osf.io/gsq5j; https://doi.org/10.17605/OSF.IO/GSQ5J).

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

## Acknowledgements

This project was funded through an Australian Research Council Future Fellowship, awarded to EP (FT190100077). The funder of the study had no role in study design, data collection, data analysis, data interpretation, or writing of this manuscript. We are grateful to Marc Stears for helpful comments on a previous draft of this manuscript.

## Author contributions

T.W.: Conceptualisation, Methodology, Investigation, Formal analysis, Writing – original draft. S.P.: Conceptualisation, Methodology, Investigation, Formal analysis, Writing – review and editing. J.A.: Conceptualisation, Methodology, Formal analysis, Writing – review and editing. H.C.: Conceptualisation, Methodology, Formal analysis, Writing – review and editing. W.L.: Funding acquisition, Conceptualisation, Methodology, Formal analysis, Writing – review and editing. T.K.: Conceptualisation, Methodology, Data Curation, Investigation, Formal analysis, Writing – review and editing E.P.: Funding acquisition, Conceptualisation, Methodology, Investigation, Formal analysis, Supervision, Writing – original draft.

## Competing interests

The authors declare no competing interests.
