## [Transparent Peer Review file · Communications Psychology]

Understanding phenomenological experiences of Autistic inertia using online community discourse

Corresponding Author: Professor Elizabeth Pellicano

Version 0:

Decision Letter:

Dear Professor Pellicano,

Thank you for your patience during the peer-review process. Your manuscript titled "'Slow to start, slow to stop": Understanding phenomenological experiences of Autistic inertia using data mining techniques" has now been seen by 2 reviewers, and I include their comments at the end of this message. They find your work of interest but raised some important points. We are interested in the possibility of publishing your study in Communications Psychology, but would like to consider your responses to these concerns and assess a revised manuscript before we make a final decision on publication.

We therefore invite you to revise and resubmit your manuscript, along with a point-by-point response to the reviewers. Please highlight all changes in the manuscript text file.

Editorially, we consider it important that the revision clarifies how the relationship between monotropism and inertia was defined and evaluated.

I am attaching an Editorial Requests Table that details critical reporting requirements for the revised manuscript. Please attend to each item and ensure your manuscript is fully compliant. If your revised manuscript is not aligned with these requests on major issues, such as those concerning statistics, it may be returned to you for further revisions without re-review.

Please submit the following items:

- Revised manuscript
- Point-by-point response to the referees' comments
- Cover letter (as a separate document)
- <https://www.nature.com/documents/nr-reporting-summary.pdf> Nature Research Reporting Summary
- Completed Editorial Request Table (attached).

via this link: Link Redacted .

**** This url links to your confidential home page and associated information about manuscripts you may have submitted or be**

reviewing for us. If you wish to forward this email to co-authors, please delete the link to your homepage first **

Additional guidance is available in our style and formatting guide Communications Psychology formatting guide.

Best regards,

Jennifer Bellingtier

Jennifer Bellingtier, PhD
Senior Editor
Communications Psychology

REVIEWER EXPERTISE:

Reviewer #1 Autism, qualitative data

Reviewer #2 Autism, qualitative data

REVIEWER REPORTS:

Reviewer #1 (Remarks to the Author):

Many thanks for the opportunity to read your interesting and timely paper!

I think there are some "room for improvement" but it is mainly food for thought/request for some more reflection.

Please see my comments in the attached document.

Main points:

1) reflect more upon impact of researchers' neurotype

2) as I view inertia as "part of" autistic cognition (part of being monotropic, so you can't cut it, you need to understand and learn to live with it/navigate in relation to it similar to how one navigate in relation to underload/overload) I would like to have a less "evaluating" way of referring to the different sides of it - either starting or stopping can be disabling/frustrating, even it may feels wonderful when you are in the monotropic high and find it hard to stop. I have no clear suggestion here, more that it got me thinking and have added a comment on this.

Reviewer #2 (Remarks to the Author):

This study examines an important under-studied phenomenon, with a larger sample size than previous studies. It is generally well-written and demonstrates understanding of the emerging (and minimal) science to date on autistic inertia. The findings herein generally corroborate and expand on current understanding. I am concerned about the methodological decision to include posts about "monotropism" and "flow state" mixed in with posts discussing autistic inertia and for these posts to be treated the same, as if the posters had identified they were talking about inertia. Perhaps these two phenomena are intertwined and interacting, however, the informants did not make this connection, nor was this reached through analysis, rather it was the researchers making this call at the start of the study. That is to say, it wasn't reddit posters making this connection or the researchers noticing that posters discussing inertia gave descriptions aligning with monotropism. Then in the findings this is presented as though these overlaps were organic and inherently within the data. This could potentially be remedied by revisiting the analysis and separating posts about monotripism from posts about inertia. Each given separate consideration and then compared across and within categories. Then the similarities, differences, and relationships between inertia and monotropism/flow state could be explored. Conceptual clarity around autistic burnout could be enhanced and be clearly delineated from inertia itself. The inclusion of "extreme states" should perhaps not be in the core definition proposed in figure 4. While extreme states may be a somewhat frequent and important experience, I don't think the case has been made to include it as part of a core definition.

Version 1:

Decision Letter:

Dear Professor Pellicano,

Your manuscript titled "'Slow to start, slow to stop": Understanding phenomenological experiences of Autistic inertia using data mining techniques" has now been seen by our reviewers, whose comments appear below. In light of their advice I am delighted to say that we are happy, in principle, to publish a suitably revised version in Communications Psychology.

We therefore invite you to revise your paper one last time to address the remaining concerns of our reviewers and a list of editorial requests. At the same time we ask that you edit your manuscript to comply with our format requirements and to maximise the accessibility and therefore the impact of your work.

EDITORIAL REQUESTS:

SUBMISSION INFORMATION:

OPEN ACCESS:

* **TRANSPARENT PEER REVIEW:** Communications Psychology uses a transparent peer review system. On author request, confidential information and data can be removed from the published reviewer reports and rebuttal letters prior to publication. If you are concerned about the release of confidential data, please let us know specifically what information you

would like to have removed. Please note that we cannot incorporate redactions for any other reasons.

* **DATA AVAILABILITY:**

Link Redacted

Best regards,

Jennifer Bellingtier

Jennifer Bellingtier, PhD
Senior Editor
Communications Psychology

REVIEWER EXPERTISE:

Reviewer #1 Autism, qualitative data

Reviewer #2 Autism, qualitative data

REVIEWERS' COMMENTS:

Reviewer #1 (Remarks to the Author):

Many thanks for the opportunity to read the revised version of your manuscript! I think it is ready to go into publication. I have no further comments other than I found it a joyfully interesting (and of course important contribution to the autism research field) read and I look forward to refer to it in my own future publications. Thank you for this work!

Reviewer #2 (Remarks to the Author):

Response from reviewer to author rebuttal:

Reviewer 2, point 7: Concern regarding treatment of data pertaining to monotropism and autistic inertia:

The explanation provided in this rebuttal letter does a good job clarifying this. The update to the manuscript (p7-8) is very brief and does not provide that clarity. I recommend adding a clearer description in the manuscript, similar to the rebuttal letter.

** Visit Nature Research's author and referees' website at <http://www.nature.com/authors>

for information about policies, services and author benefits**

Response to reviewers

Reviewer 1

1. *“Many thanks for the opportunity to read your interesting and timely paper! I think there are some ‘room for improvement’ but it is mainly food for thought/ request for some more reflection.”*

Response: We are very grateful for the reviewer’s positive comments on our manuscript, and for their very thoughtful, constructive feedback.

2. *“Main point: Reflect more upon impact of researcher’ neurotype”*

Also, in the Reviewer’s comments in the manuscript itself, they noted (on p. 9): “Was it a “insider only” group of researcher or neurodiverse group of researchers (consisting of researchers of different neurotypes). If either an insider-only or a neurodiverse group of researchers please reflect upon impact of your same or different “frame of analyses” (based on your position as either insider or outsiders to autistic experience)”

Response: Our team is a neurodiverse team consisting of Autistic advocates and researchers and non-Autistic researchers – therefore a mix of ‘insider’ and ‘outsider’ researchers. Four of the team members (three Autistic, one non-Autistic) have worked together in a partnership for the past four years, to understand the nature of Autistic inertia. All are deeply committed to, and strong advocates for, the neurodiversity paradigm, the social model of disability, and of actively including Autistic people in autism research. For this particular study, we brought three new researchers to the team (all non-autistic; one computer scientist/linguist and two students – one with an Autistic sibling), both to offer additional expertise and to build their capacity in research adopting a participatory approach. The diverse nature of our team and our participatory ways of working meant that we moved beyond neurotypical ‘frames of analysis’ (Schneid & Raz, 2020) by looking at inertia – and of Reddit users’ reported experiences of inertia – from different perspectives. The nature of our data (qualitative reports) mean that we often deferred to our Autistic partners’ experiences to try to understand Reddit users’ experiences. It is perhaps for this reason that, as one of our Autistic partner’s said on approving the final manuscript for submission, “Reading through it was very validating”. We have since added more information on positionality to the manuscript (see p. 10).

3. *“Main point: as I view inertia as “part of” autistic cognition (part of being monotropic, so you can’t cut it, you need to understand and learn to live with it/ navigate in relation to it similar to how one navigate in relation to underload/ overload) I would like to have a less “evaluating” way of referring to the different sides of it - either starting or stopping can be disabling/ frustrating, even it may feels wonderful when you are in the monotropic high and find it hard to stop. I have no clear suggestion here, more that it got me thinking and have added a comment on this.”*

The reviewer also clarified in the manuscript itself (on p. 24): “+ I do understand the ‘vicious circle of inertia’– think it is well founded. But I also want to stress that it may (from my own experience and thinking) be “normal” to autistic cognition. Currently when I read about inertia as a ‘precursor to burnout’ at the end of figure 4 I get a bit worried this may lead to an interpretation that the path forward is to ‘avoid’ inertia. I would rather think of it as ‘part of an autistic way of being’ (and an aspect of part of being monotropic) it is not something you can cut off but you can learn to cope with it and the path is therefore not to avoid it but to understand it and the “mechanism” in order to learn to cope with it

So perhaps more focus on “The cyclical nature of Autistic inertia” and not that either part is ‘better/ positive’ than the other it is just part of the cycle”

Response: We are grateful to the reviewer for these astute comments. We agree that inertia is likely to be part and parcel of autistic cognition and do not want to suggest that Autistic people should try to avoid it or seek to remediate it.

We have since amended the figure so that it reads “The cyclical nature of Autistic inertia”, as suggested (see p. 15). And we have also read through the manuscript and made changes where necessary to remove any value-laden interpretations of Autistic inertia.

Regarding burnout specifically: Although Reddit users suggested that inertia – and especially the energy depletion associated with the cycle of inertia – could make them more prone to burnout (see p. 14), we agree that the precise relationship between Autistic inertia and Autistic burnout is unclear. It could be that inertia is one precursor to burnout, as users suggested. But it could also be that burnout could exacerbate inertia. Or both. We have since removed the sentence, “Autistic inertia could be a precursor to burnout” in Figure 4, and elaborate on our discussion of this issue on p. 22, in line with Reviewer 2’s comment (see below).

4. *Comment p. 3. “Or “unknowing” what we ‘already know about autism’.”*

Response: Thank you for pointing out David Jackson-Perry’s paper, of which we were unaware. We have now read and cited it (see p. 3).

5. *“Love the figure 4/ defin: but why defined either can ‘t start or can ‘t stop as positive/negative? (couldn ‘t make a comment in the figure)*

I find both of them disabling, even if it may “feel fine” as long as you are in the hyperfocused ‘high’ some autistic people may choose to avoid seeking it as they are worried about not being able to stop until exhausted

So suggest revise “both negative (can ‘t start) and positive (can ‘t stop) consequences” perhaps just cut it and only have:

These difficulties can occur regardless of willpower, and yield different consequences.”

Response: Thank you for pointing this out. We wholeheartedly agree that both ‘can’t start’ and ‘can’t stop’ can be disabling. We have since amended the phrasing in Figure 4 so that it reads, **“These difficulties can occur regardless of willpower, and yield different consequences (some negative, some positive).”**

Reviewer 2

6. *This study examines an important under-studied phenomenon, with a larger sample size than previous studies. It is generally well-written and demonstrates understanding of the emerging (and minimal) science to date on autistic inertia. The findings herein generally corroborate and expand on current understanding.”*

Response: We thank the reviewer for their encouraging words on our manuscript, and for highlighting the important conceptual issues below.

7. *I am concerned about the methodological decision to include posts about “monotropism” and “flow state” mixed in with posts discussing autistic inertia and for these posts to be treated the same, as if the posters had identified they were talking about inertia. Perhaps these two phenomena are intertwined and interacting, however, the informants did not make this connection, nor was this reached through analysis, rather it was the researchers making this call at the start of the study. That is to say, it wasn't reddit posters making this connection or the researchers noticing that posters discussing inertia gave descriptions aligning with monotropism. Then in the findings this is presented as though these overlaps were organic and inherently within the data.*

This could potentially be remedied by revisiting the analysis and separating posts about monotripism from posts about inertia. Each given separate consideration and then compared across and within categories. Then the similarities, differences, and relationships between inertia and monotropism/flow state could be explored.

Response: Thank you. We included ‘monotropism’ and ‘flow state’ as search terms because users may have been describing inertia-like experiences without using the term ‘inertia’, simply because they were not familiar with it. We wanted our search to be as inclusive as possible in the initial dataset, so that we did not miss relevant posts that used different terms or language. Both concepts overlap with some aspects of inertia, so we wanted to capture posts where users described difficulties starting, stopping, or switching activities in those contexts. While we did analyse some posts containing those search terms, critically, these were only analysed when they described experiences consistent with inertia. For example, in one included post, a user said that their monotropism meant that they experience “heavy irritation or stress because of the small amount of effort it takes to change from one task to another”. In another example of an included post containing the term ‘flow’, one user described the ‘can’t stop’ aspect of inertia: “Okay, does anybody else have extreme trouble finding motivation to do something, but then when you finally have the motivation, you like ZERO in on it to the point where you can't get yourself to eat, sleep or shower until that thing is done or mostly done? Like is that just a flow state thing?”

Importantly, posts that discussed monotropism or flow states but were not describing inertia-like experiences were excluded from the analysis. Most of the excluded posts under ‘monotropism’ were theoretical discussions, recommendations to others to read more about the concept, or references to the ‘monotropism questionnaire’. A smaller number discussed personal experiences, but these were not related to inertia. For example, one user wrote about monotropism in relation to difficulty with perception: “Does anybody else struggle with the act of perceiving their own face in totality? It’s hard to describe, but I can only judge parts of my face one after one, leaving me unable to evaluate my face as a whole. Could it be monotropism?”

Similarly, posts mentioning ‘flow state’ were excluded when they referred to enjoyable experiences of intense focus or doing something effortlessly, without mention of difficulties initiating, stopping, or shifting activities. For example, one user describes how their work becomes intuitive and effortless during flow: “I didn’t miss a beat. I came to a solution. It was really complex DeFi analytics code. And I was startled by this high-functioning flow state. No impostor syndrome (crypto is mostly dudes on the technical side and I’m literally the only one on my team without facial hair), no weirdness, I just dove in and connected a lot of dots”.

So, rather than treating ‘monotropism’, ‘flow state’, and ‘inertia’ as inherently similar constructs, our approach was to treat the search terms as a way to find potentially relevant posts. They were included in the analysis only when the content itself aligned with inertia-related experiences, not simply because of the search term. We have since clarified this approach in the text (see pp. 7-8).

8. *Conceptual clarity around autistic burnout could be enhanced and be clearly delineated from inertia itself.*

Response: We thank the reviewer for raising this issue – which was also raised by Reviewer 1. As we noted we removed the sentence, “Autistic inertia could be a precursor to burnout” in Figure 4, and elaborated on our discussion of the relationship between autistic burnout and inertia on page 22.

9. *The inclusion of "extreme states" should perhaps not be in the core definition proposed in figure 4. While extreme states may be a somewhat frequent and important experience, I don't think the case has been made to include it as part of a core definition.*

Response: This is a good point. This experience of ‘extreme states’ has been raised in the current study and in a previous one (Rapaport et al., 2023), so for this reason, rather than remove reference to it entirely, we have tempered our wording, to suggest that it “**can be** experienced as living ‘in extremes’” (see Figure 4).

We thank the reviewers, once again, for their hugely constructive comments.

Response to reviewers

Reviewer 1

1. *“Many thanks for the opportunity to read the revised version of your manuscript! I think it is ready to go into publication. I have no further comments other than I found it a joyfully interesting (and of course important contribution to the autism research field) read and I look forward to refer to it in my own future publications. Thank you for this work!”*

Response: We are very grateful for the reviewer’s positive comments on our revised manuscript.

Reviewer 2

2. *“Concern regarding treatment of data pertaining to monotropism and autistic inertia. The explanation provided in this rebuttal letter does a good job clarifying this. The update to the manuscript (p7-8) is very brief and does not provide that clarity. I recommend adding a clearer description in the manuscript, similar to the rebuttal letter.*

Response: Thank you. We have now elaborated on this issue in the manuscript (see p. 7).

We thank the reviewers, once again, for their hugely constructive comments over the course of this review process.